# Gut Fungal Communities Are Influenced by Seasonality in Captive Baikal Teal (*Sibirionetta formosa*) and Common Teal (*Anas crecca*)

**DOI:** 10.3390/ani13182948

**Published:** 2023-09-17

**Authors:** Patthanan Sakda, Xingjia Xiang, Yuannuo Wu, Xinying Zhang, Wenbin Xu, Lizhi Zhou

**Affiliations:** 1School of Resources and Environmental Engineering, Anhui University, Hefei 230601, China; patthanan423@gmail.com (P.S.); wuyuannuo2022@163.com (Y.W.); xinyingzhang1999@outlook.com (X.Z.); 2Anhui Province Key Laboratory of Wetland Ecosystem Protection and Restoration, Anhui University, Hefei 230601, China; 3Anhui Shengjin Lake Wetland Ecology National Long-Term Scientific Research Base, Chizhou 247230, China; xwb197105@163.com

**Keywords:** animal pathogen, Baikal teal, common teal, gut fungal community, saprotrophs

## Abstract

**Simple Summary:**

The fungi community has received comparatively little attention compared to bacteria in previous studies on the avian gut microbiome. Even though fungi constitute a smaller proportion of the gastrointestinal (GIT) microbiota, changes in the diversity and composition of the gut microbial population have been associated with a number of diseases. This study, concentrating on captive migratory ducks, provides light on the gut fungal population, identifies potential animal pathogens and plant saprotrophs, and reveals that seasonal variations shape the dynamics of the gut fungal community more significantly than the host species.

**Abstract:**

Understanding the dynamics of avian gut fungal communities and potentially pathogenic species across different seasons is crucial for assessing their health and ecological interactions. In this study, high-throughput sequencing was employed to examine the changes in gut fungal communities and the presence of potential pathogens between different seasons in captive Baikal teal and common teal. Between the summer and autumn seasons, both duck species showed significant differences in fungal diversity and community composition. A higher fungal diversity in both species was exhibited in the summer than in the autumn. Ascomycota and Basidiomycota were the two most common phyla, with a greater proportion of Ascomycota than Basidiomycota in both duck species in the summer. Interestingly, our study also identified animal pathogens and plant saprotrophs in the gut fungal communities. Seasonal variation had an effect on the diversity and abundance of both animal pathogens and saprotrophs. Specifically, during the summer season, the diversity and relative abundance were higher compared to the autumn season. In addition, there were differences between duck species in terms of animal pathogens, while no significant differences were observed in saprotrophs. Overall, the communities of the gut fungi, animal pathogens, and saprotrophs were found to be influenced by seasonal changes rather than host species. Therefore, seasonal variations might dominate over host genetics in shaping the gut microbiota of captive Baikal teal and common teal. This study underscores the importance of incorporating an understanding of seasonal dynamics and potential pathogens within the gut microbiota of captive ducks. Such considerations have the potential to drive progress in the development of sustainable and economically viable farming practices.

## 1. Introduction

The gastrointestinal (GIT) microbiota is a complex and varied ecosystem composed of bacteria, viruses, parasites, fungi, and other microorganisms that are essential for keeping their hosts healthy [1]. The host’s genetics, age, and the surrounding environment are all factors that affect their microbial functions [2]. Since the first identification of a diverse bacterial population in Canada geese (*Branta canadensis*) and whistling swans (*Cygnus columbianus columbianus*) [3], the majority of studies on the waterfowl’s gut microbiota have focused on bacteria [4,5,6,7]. Furthermore, recent investigations have more concentrated on the gut fungal community in wild waterbird species [8,9,10]. The intestinal fungal community composition of migratory waterbirds may fluctuate due to shifting diets and seasons [9,11], as well as a variety of habitats and captivity conditions [12]. Although fungi only make up 0.1% of the GIT microbiota [13], several medical conditions, such as inflammatory bowel disease (IBD), obesity, and allergy problems, have been associated with modifications in the richness and composition of the gut fungal community [14,15,16].

The Baikal teal (*Sibirionetta formosa*) and common teal (*Anas crecca*) are small dabbling ducks currently classified as Least Concern (LC) by the IUCN and BirdLife International. It is known that the Baikal teal breeds in eastern and northern Siberia and spends its winters mostly in China, South Korea, and Japan [17]. Meanwhile, one of the species of dabbling ducks with the greatest abundance is the common teal, which is acknowledged on a global scale [18]. It migrates throughout certain seasons between breeding sites and wintering areas, which are dispersed throughout Eurasia and North America [19]. Both Baikal teal and common teal consume a wide variety of plants, insects, and aquatic invertebrates in their diets. They have been seen consuming aquatic plants, grasses, sedges, grains, and even grains of rice and wheat [20,21]. It has been determined that the migratory common teal has a key role in the spread of Highly Pathogenic Avian Influenza (HPAI) viruses in the environment [22], which have been linked to outbreaks in many countries, including the United States [23], Germany [24], and Japan [25].

Previous studies on avian gut microbiota have primarily focused on the bacteria community, with limited attention given to fungi, despite their potential importance, and dysbiosis or imbalances in the gut fungi composition have been linked to a number of medical disorders [26]. However, there has recently been a rise in curiosity about the function of the gut fungal community [27]. To the best of our knowledge, no previous study has investigated the gut fungal community of these duck species. The objective of this study is to investigate the gut fungal community and identify potential animal pathogens and plant saprotrophs between two different seasons in captive Baikal teal and common teal in Shengjin Lake, Anhui Province, China. By examining the dynamics of the gut microbiota in these captive migratory ducks, we aim to gain novel insights into the influence of seasonal factors and host species on fungal communities, including potential pathogenic communities. Comparing our findings with previous studies on gut microbiota in wild migratory birds will help us comprehend the basic concepts and practical aspects of microbial community structure.

## 2. Materials and Methods

### 2.1. Overview of the Study Area

The location of this research was Shengjin Lake, a shallow lake with a river connection that flows into the Yangtze River (30.15–30.30° N, 116.55–117.15° E). The lake is an essential stopping and wintering habitat for migratory birds migrating through the East Asia–Australasian flyway, making it a wetland of worldwide significance. The average annual temperature is 16.14 °C, and there is 1600 mm of precipitation on average per year. During the summer, the average high temperature recorded in 2022 was 31.11 °C, the average low was 25.56 °C, and the average precipitation was 82.26 mm. While in the autumn, the average high temperature was 21.67 °C, the average low was 14.44 °C, and the average precipitation was 15.26 mm. The dabbling ducks from a captive breeding population were in separate cages for research and conservation purposes. The housing replicated their natural habitat, providing water access and sufficient space. The ducks’ diet was carefully balanced with paddy, seeds, crustaceans, and aquatic plants for proper nutrition.

### 2.2. Sample Collecting

A comprehensive collection of feces samples was carried out on captive Baikal teal and common teal, encompassing both the summer (July) and autumn (October) seasons in 2022. A total of 68 fecal samples were obtained, with each season consisting of 15 common teals and 19 Baikal teals. To ensure sample integrity, fresh fecal samples were carefully collected, ensuring no contact with soil, and promptly placed in individual labeled standard sterilized sampling bags indicating the specific duck species and date of collection. Following collection, the samples were transported to the laboratory at the School of Resources and Environmental Engineering, Anhui University, where they were stored at a temperature of −80 °C for subsequent processing.

### 2.3. DNA Extraction, PCR Amplification, and Sequencing

In total, 68 DNA samples were extracted using the SPINeasy DNA Kit for Feces (MP Biomedicals, Santa Ana, CA, USA) in accordance with the manufacturer’s instructions. After genomic DNA extraction, NanoDrop ND-1000 (Thermo Scientific, Chino, CA, USA) was used to quantify the extracted DNA. Extracted DNA was sent to the Hefei Baisheng Science & Technology Development for library construction, quantitation, pooling, and sequencing. Fungi present in the samples were identified through ITS rRNA sequencing. For fungal identification, the ITS hypervariable regions were amplified with the following primers: ITS1F (5′-CTTGGTCATTTAGAGGAAGTAA-3′) and ITS2R (5′-GCTGCGTTCTTCATCGATGC-3′).

The following protocol was used to run the PCRs on an Applied Biosystems GeneAmp 9700 thermocycler: denaturation at 95 °C for 3 min, then 27 cycles of 95 °C for 30 s, 55 °C for 30 s, and 72 °C for 45 s, followed by a final extension at 72 °C for 10 min. After PCR amplification, we used a Qubit 3.0 fluorometer to qualify each PCR product before next-generation sequencing. The Illumina MiSeq PE250 platform (Illumina, San Diego, CA, USA) was utilized in this study.

### 2.4. Bioinformatic Processing and Analysis of the Sequencing Data

The paired-end reads from sequencing were merged using Fast Length Adjustment of SHort reads (FLASH) [28]. The resulting sequences that fulfilled at least one of the following criteria were removed with the split_libraries_fastq.py script from QIIME 1.9.1: average quality score lower than 30 and containing unresolved nucleotides [29]. Quality filtering obtains high-quality clean sequences. Sequences were grouped into operational taxonomic units (OTUs). Singletons were filtered for downstream analysis. The chimeras were removed with the UCHIME algorithm (v4.2.40) [30]. The UNITE database was used to classify OTUs at the taxonomic level [31]. Fungal community comparisons were performed at the same surveying effort, using the lowest number of sequences by random selection (10,300) sequences per sample. The rarefaction curve, alpha, and beta diversity were determined using the QIIME 1.9.1 script alpha_rarefaction.py. In addition, the identification of fungal functional groups was performed utilizing the FUNGuild software package [32] based on the OTU table result. The community and relative abundance of animal pathogens and saprotrophs were analyzed separately.

### 2.5. Statistical Analysis

Four indices (Chao1, Observed Species, Shannon, and Simpson) were used in the study of alpha diversity to examine the complexity of the species diversity in a sample. The Chao 1 and Observed Species indices measure the number of distinct species in each sample. The Shannon Diversity Index evaluates the species diversity within a community, while Simpson’s Diversity Index considers the richness of species present and their distribution. All alpha indices and beta diversity were analyzed in QIIME1, and visualization was performed in R software (v 4.2.2). The Venn diagram was constructed based on the OTU with the Venn diagram package showing the number of OTUs shared and differences between groups. Non-metric multidimensional scaling (NMDS) based on OTU level was applied to show the composition differences of OTUs between groups using the vegan and ggplot2 package in the R package [33]. The linear discriminant analysis (LDA) effect size (LEfSe) analyses were performed on the website http://huttenhower.sph.harvard.edu/galaxy (accessed on 30 June 2023) [34] to compare order between groups to determine the differential abundance taxonomic features. The one-way ANOVA with Duncan’s post hoc test was used to consider the difference in alpha diversity and relative abundance of fungal composition between sample groups. Differences with a *p*-value < 0.05 were considered significant. Plots of differential alpha diversity and relative abundance were generated using the ggplot2 package in R.

## 3. Results

### 3.1. General Characteristics of the Gut Fungal Sequences

In the analysis of the samples, a total of 5,031,421 quality-filtered fungal sequences were obtained. A total of 1880 distinct fungal OTUs were identified, with the number of OTUs varying from 71 to 413 across all samples. Among these OTUs, 437 (23.24% of the total) were found to be shared for all four groups studied: common teal in autumn (CMA), common teal in summer (CMS), Baikal teal in autumn (BKA), and Baikal teal in summer (BKS). The unique gut fungal OTUs for each group were as follows: 98 for CMA, 240 for CMS, 134 for BKA, and 273 for BKS (Appendix A, Appendix A).

### 3.2. Fungal Diversity and Community Composition

Following the analysis of fungal diversity, we obtained the Chao1 and Observed species for the gut fungal samples. The results revealed variations in diversity between the two seasons in both duck species. Specifically, they were significantly higher in both Chao1 and Observed species indices in summer compared to autumn in both duck species. Notably, Baikal teal exhibited higher diversity than common teal during the autumn season, whereas no significant difference was observed during summer (Figure 1A,B).

A non-metric multidimensional scaling (NMDS) analysis was performed to investigate the inter-individual differences between groups. The main objective of the NMDS analysis was to examine the separation and clustering patterns of samples. The results revealed a clear separation between the two seasons, indicating that the fungal community composition varied significantly between summer and autumn for both hosts. However, no distinct separation was observed between the duck species (Figure 2).

During summer and autumn, the dominant fungal phyla observed were Ascomycota (81.90%), Basidiomycota (15.65%), Chytridiomycota (1.89%), and Zygomycota (0.56%) in Baikal teal and common teal. In summer, both ducks exhibited a higher relative abundance of Ascomycota, Chytridiomycota, and Zygomycota compared to autumn, while Basidiomycota was lower in summer than in autumn. Notably, during each season, the abundance of Ascomycota in common teal was higher than in Baikal teal, whereas Baikal teal had higher abundances of Basidiomycota, Chytridiomycota, and Zygomycota compared to common teal (Appendix A).

In order to investigate the detailed differences in gut fungal biomarkers between the two seasons in the two duck species, LEfSe analysis was conducted at the phylum to family level. This analysis aimed to identify potential biomarkers in the four identified groups. Using a logarithmic LDA (linear discriminant analysis) size effect value of 2.0, a total of thirty-eight fungal assemblages were identified as significant (Figure 3A). The cladogram visualization displayed distinct patterns. The family Incertaesedis was found to be more abundant in the CMA group, while the family Arthrodermataceae was more prevalent in the BKA group. At the order level, Onygenales, Saccharomycetales, Glomerellales, and Xylariales were more abundant in the CMS group. The orders Helotiales, Sordariales, Incertaesedis, Mortierellales, and Mortierellales were found to be more abundant in the BMS group (Figure 3B).

At the genus level, 14 genera exhibited a relative abundance of more than 1% (Appendix A). The dominant genera included *Cryptococcus* (21.14%), *Aspergillus* (20.23%), *Gibberella* (11.77%), and *Candida* (11.00%). Specifically, *Aspergillus* exhibited higher abundance during summer in both duck populations and was not different between duck species within the same season, whereas *Alternaria* and *Cryptococcus* displayed greater abundance during autumn compared to summer, and no difference was observed between duck species within the same season. *Cystobasidium* and *Davidiella* were significantly higher in Baikal teal in autumn, and no difference was found in both duck species in summer and common teal in autumn (Figure 4).

### 3.3. Potential Animal Pathogens and Saprotrophs

The FUNGuild analysis was employed to identify potential animal pathogens and plant saprotrophs between summer and autumn in both Baikal teal and common teal. A total of 1434 OTUs were assigned to the functional fungi, 122 OTUs (8.51%) were identified as animal pathogens (Figure 5A), and 400 OTUs (27.89%) were designated as saprotrophs (Figure 5B). For both host species, there were more unique OTUs of animal pathogens and saprotrophs in the summer than in the autumn. While common teal had more unique animal pathogen OTUs in both seasons, Baikal teal had more distinct saprotrophs than common teal.

The NMDS analysis provided insights into the structure of the community compositions of potential animal pathogens and saprotrophs across the samples. The analysis revealed that the animal pathogen and saprotroph fungi displayed separated clusters between the two seasons and similar variations between the two host species. The NMDS analysis for animal pathogenic fungi resulted in a stress value of 0.2196, and the NMDS analysis for saprotrophs yielded a stress value of 0.1689 (Figure 6).

The study employed a non-parametric test to evaluate the diversity and relative abundance of animal pathogens and saprotrophs in the two duck species across the two seasons. The findings indicated significant variations in the diversity and relative abundance of animal pathogens and saprotrophs between the two seasons in both duck species, with higher diversity and relative abundance recorded during summer compared to autumn and no difference discovered between the two duck species in the same season (Figure 7A,B). 

Among the animal pathogens, genus *Candida* had the most abundance across all groups with the significant highest in the common teal during the summer. According to the *Candida* panel shown in Figure 4, it seems a few common teals acquire candidiasis throughout the summer. This may have affected the prevalence of *Candida* in this group, which would have subsequently affected the overall results. *Candida* consisted of several species, including *C. albicans*, *C. athensensis*, *C. tropicalis*, *C. friedrichii*, and *C. metapsilosis* (Appendix A). Diverse genera of saprotrophs were found to vary across the two seasons in both duck species. For example, genera Catenaria and *Wickerhamomyces* were exclusively detected during the summer season. On the other hand, the genera *Acrocalymma*, *Lasiosphaeriaceae*, *Lophiostoma*, *Ophiosphaerella*, *Paraconiothyrium*, *Paraphaeosphaeria*, and *Rhizophlycti* were detected in the autumn season. However, the genera *Davidiella*, *Debaryomyces*, *Guehomyces*, *Penicillium*, *Rhodosporidium*, and *Talaromyces* were observed in all groups throughout the study (Appendix A).

## 4. Discussion

Investigating the microbiota of captive birds is of great importance due to its potential impact on bird health, welfare, and conservation efforts. The gut microbiota plays a crucial role in digestion, nutrient metabolism, immune modulation, and overall host well-being [35]. Captive environments can significantly influence the gut microbiota of birds such as phylogenetic status, diet changes, treatments, and reduced contact with other individuals leading to distinct microbial communities [36,37].

The results of this study showed that in captive Baikal teal and common teal, the composition and alpha diversity of the intestinal fungal communities varied significantly across two seasons. These findings are in accordance with other studies that found seasonal fluctuation in the gut microbiota in several avian species, such as Swainson’s thrushes and gray catbirds [38,39]. There are several reasons for the seasonal fluctuation in gut fungal communities. The availability of food supplies is one important element. There are variations in the variety and amount of food sources throughout the year, such as the switch from an insect-based diet to plant-based matter [40]. In contrast, to maintain stability in the food fed to captive ducks in our study, we applied nutritional control across the seasons. This strategy tried to reduce how much different diets affected the outcomes. The composition of the gut fungal community can be directly impacted by these variations in food availability since various fungal species may be favored or inhibited based on their individual nutritional preferences. The alpha diversity in the gut fungal communities in the two hosts differed significantly between summer and autumn, potentially due to the high temperature inducing remarkable taxonomic changes in the gut microbiome of ducks [41]. Significant variations were indicated by observable shifts in the abundance and diversity of fungal communities between seasons. These changes can be ascribed to seasonal changes in climatic variables, which are probably connected to shifts in the observed fungi composition [42]. There has been a thorough analysis of environmental cues that influence the sustenance of fungus, such as gases, light, stress, temperature, pH, and humidity, which may favor the growth and proliferation of certain fungal species over others [43,44]. Therefore, fungal microbiomes in captive Baikal teal and common teal without affecting different diets are characterized by their dynamic nature and natural changes. 

Overall, the fungal communities in both the Baikal teal and common teal were dominated by Ascomycota, Basidiomycota, and Chytridiomycota across different seasons. However, we found seasonal fluctuation in the abundance of these phyla. Ascomycota and Chytridiomycota showed a significant increase throughout the summer, whereas Basidiomycota exhibited a higher level in autumn. Apart from the Ascomycota, we observed seasonal variation in abundance in a variety of taxa. In particular, the abundance of the endophyte *Acremonium* species, as well as *Aspergillus* and *Debaryomyces*, increased in the summer, in concordance with a previous study of the seasonal diversity of endophytic fungi on medicinal plants [45]. *Aspergillus* exhibited a seasonal pattern, with a higher number in the hot and dry months. It could be caused by tiny spherical spores, which during dry periods can spread quickly through the air, but during times of rain, the spores may settle with the rain [46]. The differences in the occurrence of endophytic fungi and their colonization frequency were caused by climatic conditions such as temperature, rainfall, and air humidity [47]. Thus, the prevailing climatic conditions throughout the several seasons have an impact on the abundance of the predominant fungi phyla.

Ascomycota and Basidiomycota communities are dominantly similar to wild birds [48], wintering hooded cranes, domestic geese, and lesser white-fronted geese [8,9,10]. However, Ascomycota and Basidiomycota were revealed to be increased in inflammatory bowel disease (IBD) [49]. In addition to playing a crucial part in the physiology, development, and metabolic processes of host species, the Ascomycota phylum of fungi also produces numerous essential enzymes that aid in the digestion of complex carbohydrates [50,51]. However, Basidiomycota is another diverse fungal phylum with a wide range of ecological roles, including decomposition, mycorrhizal associations, and plant pathogenesis [52,53]. The potential functional roles of Ascomycota and Basidiomycota in avian gut health are not well understood. However, prior research has revealed that they may be mostly connected to ducks primarily consuming plant roots and leaves. These fungi may aid hosts in digesting and absorbing a greater number of plant saprotrophs [9,10].

*Cryptococcus*, a yeast-like fungal genus, has been identified as the predominant species in the fungal community inhabiting the intestines of captive Baikal teal and common teal. These fungi are well known for causing Cryptococcosis, now recognized as one of the deadliest fungal diseases worldwide and a highly significant fungal disease that affects animals worldwide. It may infect a wide variety of animals, including humans, as well as occasionally birds, reptiles, and amphibians. *Cryptococcus* yeasts are widely distributed in nature and are commonly associated with avian droppings and decaying wood [54,55]. Along with *Cryptococcus*, *Aspergillus* is another well-known genus of filamentous fungi that is frequently found in a variety of environments. Most *Aspergillus* species are benign, but a few of them have the ability to create mycotoxins, which may be exceedingly harmful to both people and animals [56]. Since they were first discovered many years ago, *Aspergillus* infections in birds continue to be a leading cause of death in both captive and occasionally free-living birds [57]. Additionally, other prevalent intestinal fungi, such as *Gibberella*, are frequently linked to plant and soil illnesses [58]. For instance, the *Fusarium pseudonygamai* regularly affects important cereal crops like maize and sorghum and causes plant diseases [59]. 

In this study, potential animal pathogens and saprotrophic fungi in both Baikal teal and common teal during the summer and autumn seasons were identified by FUNGuild analysis. Our results showed that seasonal fluctuations affected the diversity and abundance of animal pathogens, with noticeably greater levels seen throughout the summer, corresponding with the general trends of the fungi community pattern, in contrast to the previous study on migratory hooded cranes, which observed a higher relative abundance of potential animal pathogens during the late winter period [11]. Climate changes that impact pathogen survival outside of hosts may be responsible for these seasonal fluctuations and may alter host behaviors, which may interact with other elements [60]. Additionally, seasonal variations in the host immune system may have an impact on the infections’ ability to proliferate within hosts [61]. However, there were differences in the diversity and abundance of animal pathogens between the two duck species. Common teal had a significantly higher abundance than Baikal teal. According to Fisher et al. [62], fungi display a great degree of diversity and are widely acknowledged as diseases across a variety of host taxa. In order to establish stable circumstances for their life, fungi are able to identify particular host tissues that serve as an attractant [63]. In recent years, the epidemiological patterns of some fungal diseases associated with domesticated and wild animals have changed, showing an increased prevalence, death toll, or change in populations at risk [64]. The main causes of disease in poultry are either tissue invasion and damage or the production of toxins in grain or finished feeds, which are ingested by the host animal, causing mycotoxicosis [65]. Among the animal pathogens in captive Baikal teal and common teal, the *Candida genus* (*Candida albicans*, *C. athensensis*, *C. tropicalis*) exhibited the highest relative abundance, which is considered a medically important species [66]. The sixteen *Candida* species were obtained from captive parrots in Italy [67]. In wild animals, *Candida* aligns with previous findings indicating the potential pathogenicity of *Candida* species and their substantial role in the epidemiology of various mycotic diseases. This association has been reported in diverse wild avian species, including woodcocks, coots, bean geese [68], broilers [69], Mallard ducks [70], and pigeons [71].

Additionally, we found that summer had a higher diversity and abundance of plant saprotrophs than the autumn season. Compared to the previous study, the number of wintering hooded cranes was higher in the late winter season than in the early and middle seasons, which may relate to the rapid digests of food including freshwater aquatic plants at that time [11]. While there was no difference between the two species of duck, seasonal changes in environmental parameters have a significant impact on the saprotrophic fungus communities. In spite of this, the predominant genus *Davidiella* was extensively distributed in a variety of proportions in soils, aquatic environments, and plants [10], and *Davidiella tassiana* was one of several harmful fungal endophytes in Aleppo pine (*Pinus halepensis*) [72]. However, the prominent genus *Catenaria* has been identified as a crucial zoosporic fungus that exhibits reproductive reliance on a single host and can naturally control populations of dipteran insects or nematodes [73].

The sharing of gut fungal communities concluding the animal pathogenic fungi of the mixed captive Baikal teal and common teal, which coexist in the same diet and environment, likely emerges from shared space utilization and high interactions. The insufficient separation between various host types increases the chance of symbionts including disease development more than moderate mixing does [74]. The general pathogen transmission involves a host’s excretion, survival in an environment outside of their body, consumption, and subsequent colonization of a new host [75]. Nevertheless, adaptation and survival in new hosts will depend on their inherent chemical, physiological, or behavioral characteristics [76]. Earlier research on wild waterfowl demonstrates notable variations in fungal communities and animal pathogen levels between different host species with overlapping foraging but consuming various diets [9]. However, the proximity of foraging sites makes the transmission of gut microbiota among different hosts easier [6]. Hence, in addition to environmental variations, the greater similarity of fungal communities and pathogenic fungi was shaped by increased contact, overlapping feed patterns, and comparable captive conditions among different hosts.

## 5. Conclusions

Finally, studying the gut microbiota of confined birds is essential for comprehending their well-being, health, and conservation efforts. Seasonal variations in the structure and function of the gut fungal communities can be attributed to dietary habits, living arrangements, and disease exposure in Baikal teal and common teal. Ascomycota and Basidiomycota were the two most prevalent phyla found in the gut fungi of these bird species, and they may have a functional role in digestion, nutrition metabolism, and immunological regulation. More importantly, Ascomycota and Basidiomycota are very abundant phyla in the environment, which would reflect their abundance in the duck gut as well. Both duck species had higher levels of unique fungal OTUs and alpha diversity in the summer compared to autumn, but Baikal teal showed higher levels than common teal within a season (autumn), suggesting seasonal changes in the gut fungal community rather than host species. Using the FUNGuild analysis, the presence of saprotrophic fungal communities and animal pathogens was discovered. This analysis revealed interesting and significant animal pathogens like *Candida* with an intriguing number that have been reported in avian infections, highlighting the need for additional research and effective management techniques. Our knowledge of captive birds’ risk of disease susceptibility is influenced by our understanding of the dynamics of their intestinal fungal populations. Additionally, it emphasizes how crucial it is to manage and preserve the population health of caged birds while taking seasonal fluctuations and the surrounding living environment into account. In order to maintain the ideal gut microbiota composition in captive bird species, it is necessary to conduct more studies to examine the precise relationships between gut fungal populations and host health.

## Figures and Tables

**Figure 1 animals-13-02948-f001:**
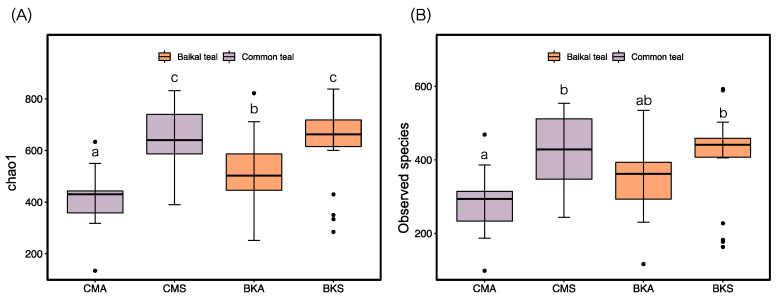
The alpha diversity indices: Chao1 (**A**) and Observed species (**B**) were compared among the CMA, CMS, BKA, and BKS groups. Significant differences at *p* < 0.05 are denoted by lowercase letters and the black dots show the extreme values.

**Figure 2 animals-13-02948-f002:**
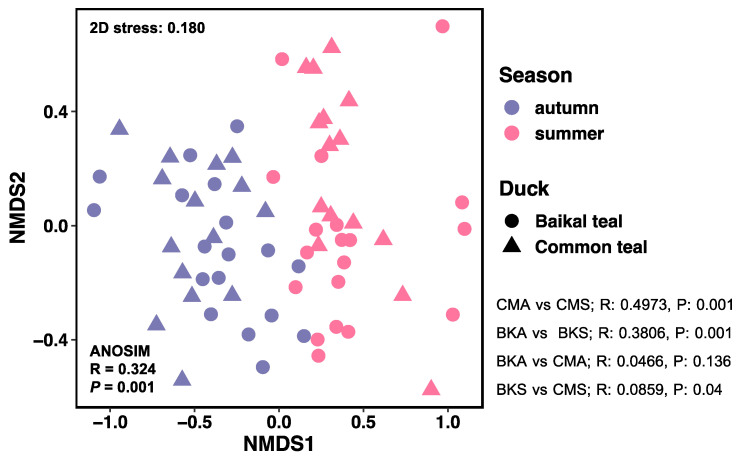
Non-metric multidimensional scaling (NMDS) plot using Bray–Curtis dissimilarity distances for fungal community composition between samples from two host species across two seasons (stress = 0.180). Samples from autumn are denoted by the color purple, while samples from summer are represented by the color pink. The host species are distinguished by the following symbols: circles for Baikal teal and triangles for common teal.

**Figure 3 animals-13-02948-f003:**
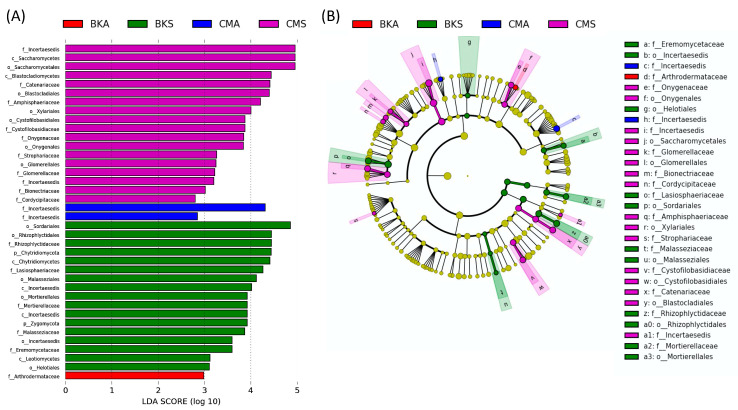
The analysis of linear discriminant analysis (LDA) effect size (LefSe). Significant results were determined based on an effect size greater than 2.0 and an alpha value less than 0.05. The LDA scores are represented as a histogram, showcasing the differentially abundant gut fungal taxonomy at the family level among the four groups. The size of each bar depicts the effect size of specific taxa within the respective group (**A**). A cladogram identifies taxonomically consistent differences between the fungal community members of the four groups, highlighting distinct lineages and their corresponding effect sizes (**B**).

**Figure 4 animals-13-02948-f004:**
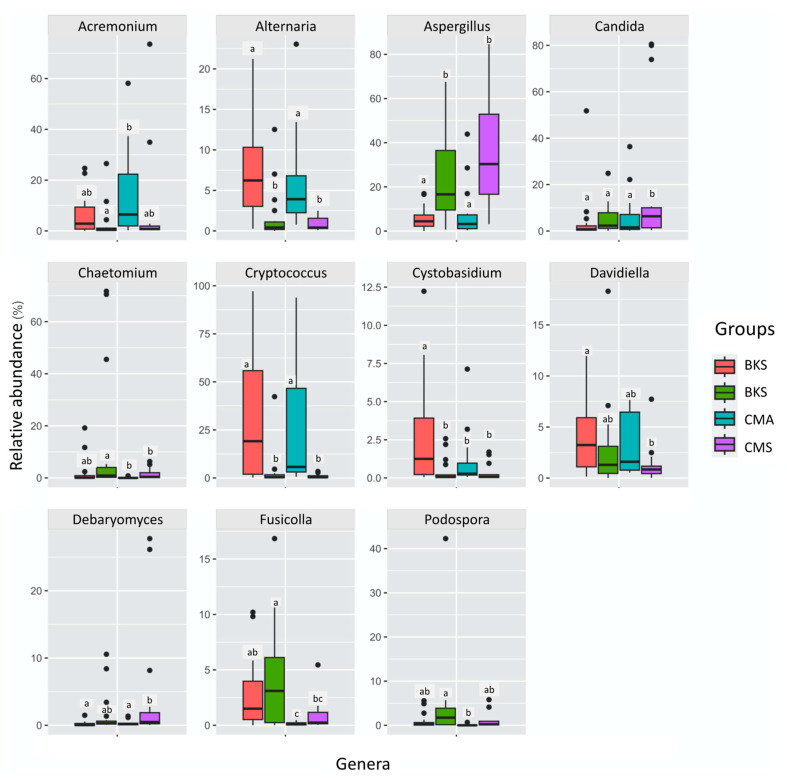
One-way ANOVA with Duncan’s post hoc test showed significant differences in relative abundance across groups at the genus level. Small letters are used to indicate significant differences at *p* < 0.05. The whiskers extended from the box to the minimum and maximum values and the black dots show the extreme values.

**Figure 5 animals-13-02948-f005:**
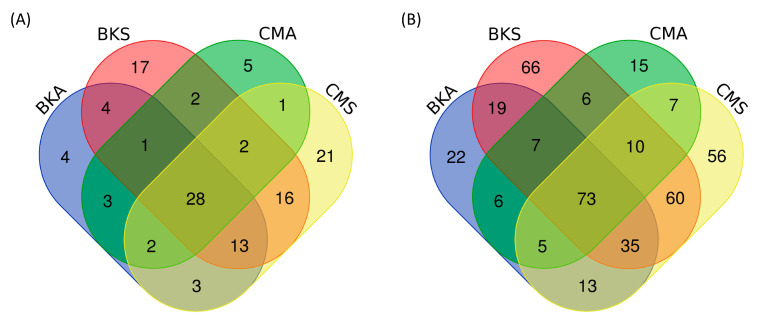
The Venn diagram showcases the number of common and unique OTUs of potential pathogen fungi across the four groups. Specifically, it presents the overlapping and distinct OTUs for animal pathogens (**A**) and saprotrophs (**B**).

**Figure 6 animals-13-02948-f006:**
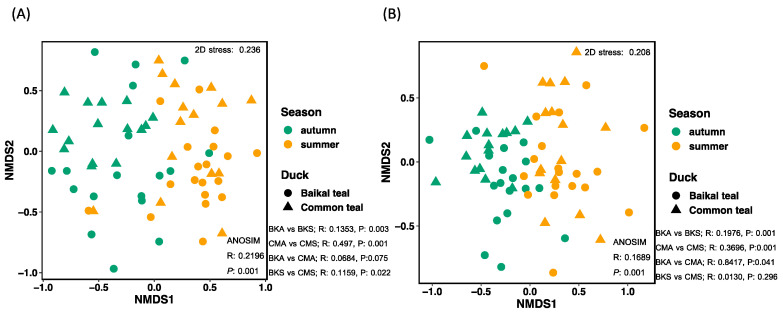
Non-metric multidimensional scaling (NMDS) plot using Bray–Curtis dissimilarity distance insights into the association of (**A**) animal pathogens and (**B**) saprotrophs in two seasons across two host species. In both ordination plots, samples from different seasons are differentiated by color, and the two host species are differentiated by different symbols.

**Figure 7 animals-13-02948-f007:**
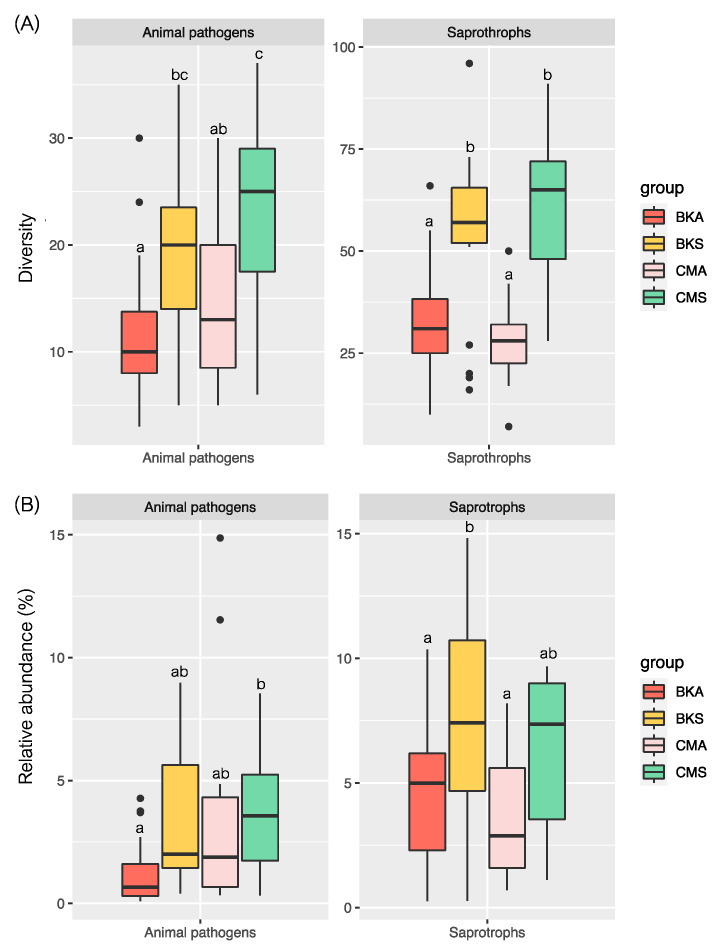
The diversity and relative abundance of animal pathogen and saprotroph fungal communities between two seasons in two duck species; (**A**) diversity and (**B**) relative abundance, based on the non-parametric with Duncan test. Significant differences at *p* < 0.05 are denoted by lowercase letters and the black dots show the extreme values.

## Data Availability

The raw data have been submitted to the NCBI Sequence Read Archive (BioProject identifier (ID): PRJNA1003563).

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
