# Peer review of "Gut Fungal Communities Are Influenced by Seasonality in Captive Baikal Teal (Sibirionetta formosa) and Common Teal (Anas crecca)"

_animals, 2023, doi:10.3390/ani13182948_

Round 1

Reviewer 1 Report

In this paper, the authors compare the fungal communities in the gut of two different duck species during summer and fall. Unfortunately, mycology is an undervalued portion of microbiome studies, and I greatly appreciate the authors work and how nicely they described their findings. I think this paper is a great addition to the scientific community that works with microbiology, avian medicine, and wild bird species.

 I like how the authors displayed their results in a very straight-forward way, without losing any valuable information. They used statistics tools that resulted in clear and simple data, which facilitates the reading. Given how direct the manuscript is, I find the title is too long. I suggest something like “Gut fungal communities are influenced by seasonality in captive ducks”.

My main comments about this paper refer to the material and methods. The sampling methodology in this specific research project is fundamental since fungi are an enormous environmental contaminant, and this method needs to be very meticulously explained. In addition, the meteorology information of the two seasons studied would have been appreciated.

Another issue is about the fungal pathogens mentioned in this paper. It seems like Candida was a problem in a few common teal ducks during summer, but it may not be an accurate representation of the whole group, which could potentially bias this small portion of the analysis. 

Some more specific comments below:

Lines 44-46: This first part of this sentence is too general to cite only 3 papers. There are countless papers on the gut bacterial microbiota of birds. Perhaps you would want to rephrase this to “waterfowl” instead of “birds”, but you still should add more references (perhaps some of the first bacterial microbiome studies in waterfowl).

Lines 85-87: Providing the average annual temperature and precipitation is not enough. It would be important to report the average high and low temperatures and precipitation during summer and fall of 2020 since these factors would highly impact the fungi proliferation and abundance of specific genera.

Lines 92-100: I wish the sample collection was more detailed. How were the feces retrieved without touching the soil, did you use a cloacal swab? How much feces were collected in each Ziploc bag? Were these bags sterile (like Whirl-Pak or similar grade bags)? What was the volume of feces collected from each bird?

Lines 104-105: I am not sure why the authors did electrophoresis of the extracted DNA before the PCR amplification.

Line 122: Please clarify what the OTU acronym means.

Lines 153-154: There is a typo here. According to Figure S1, there are 134 unique OTUs for the BKA group.

Lines 242-245: Considering the Candida panel of Figure 4, it seems that there were a few common teals going through candidiasis during the summer. This potentially influenced the abundance of Candida in this group, affecting the overall results (as seen in Figure S3).

Lines 380-383: More importantly, Ascomycota and Basidiomycota are very abundant phyla in the environment, which would reflect their abundance in the bird gut as well.

Figure 3: This figure is missing. I don’t know what Figure 3 looks like, but I really appreciate Figure S3 and I think it should be in the paper, except if its content overlaps with that of Figure 3.

Figure 4: Could you please clarify what the black dots represent, if the y axis shows relative abundance? What are the whiskers from the boxplot representing?

Table S3: This table is unnecessary since this information is displayed in Figure 6.

There are very minor language mistakes that can easily be corrected by the copyediting team.

Author Response

We hope this letter finds you well. I am writing to submit the revised version of our manuscript titled “Seasonal variations over host species in shaping gut fungal and potential pathogenic communities in mixed captive Baikal Teal and Common Teal” which was initially submitted to “Animals” under the manuscript ID animals-2573308. We would like to express our gratitude to you for the insightful feedback and valuable suggestions provided during the review process.
We have carefully addressed each of the your comments and suggestions in this revised version. Please find the response file in the attachment

Reviewer 2 Report

            The intestinal microbiome contains bacteria, viruses, and mycotic organisms.  The greatest research attention has been paid to the bacteria and diverse roles in physiology and pathophysiology have been identified for many bacteria in the intestinal microbiome.  Any information on viral or mycotic organisms including identification and roles in health and disease of the host organism is interesting and important.  This research group has a publication record in the intestinal mycobiome, investigating and identifying species and some investigation of the forces they drive composition of this mycobiome.  Their work is also novel that avian species are studied.  Experimentation in avian species is more challenging as to set up experimental models in environments is not straightforward.   In some cases, as wild migratory birds, only limited data can be obtained, experimental subjects cannot be controlled.   I appreciated the methods of stool collection, this gives greater power to the model as each individual may be used as their own control rather than solely use of non-individual identified samples for a population analysis.  Therefore the present studies are interesting and important, but the authors must be more conservative in their interpretation.  A Google and Google scholar search were performed to investigate ‘intestinal fungi/mold/yeast in avian health.”  This yielded some, but not many results as the authors know.  Indeed, in human and rodent animal models, not a great deal  regarding the intestinal mycobiome is investigated and known about intestinal mycotic organisms and health.  Similar to humans, Candida is an identified pathogen for birds as well as Aspergillus.  There are others and this information on the role of intestinal mycotic organisms in health and disease needs to be presented.  The authors text is vague and incomplete.  References 6-8 are claimed to have health information, but they only make similar claims and use the word pathogen too often without any strong data relating to mycobiome and some health condition.  The authors likely know that nearly all of the literature on fungal infections is birds is pet birds, a condition which can be manipulated for experimental study.  Control of experimental conditions and variables and subjects could also be done for agricultural use of birds, such as those for the present studies.  This would be great to have and enhance the studies greatly. 

English usage and syntax mostly correct, but there are a number of areas for resolution and clarity improvement

Author Response

(The authors gave the same response as above.)

Reviewer 3 Report

The present study revealed the diversity and abundance of gut fungal flora of captive Baikal teal and Common teal in different seasons. This study provides ideas on the relationship between gut fungal flora and host health in captive poultry, but there are still some issues in this study that need further confirmation.

I have only a few minor suggestions for the authors to consider:

1. Line 176: Figure 2 and Figure 6 do not indicate the method of analysis.

2. Figure 3 is not visible in the text.

3. Lines 198-199: The expression "Figure 4" in this sentence should be aligned with the preceding expression.

4. Lines 196-202: only describes the variability of the genus in terms of seasons, and does not explain how it differs from duck to duck.

5. Figure 6: In the graphic description, there is more writing (B) in the last sentence.

6. Do Ascomycota and Basidiomycota play a role in digestion, nutrient metabolism, and immunomodulation, and can they be added as microbial agents to the feed to regulate the animal's gut fungal?

7. 10. What is the reason why Baikal teal show a higher diversity of gut fungal than Common teal?

8. This study analyzed the gut fungal flora of two different ducks, what was the reason for the selection of the Baikal teal, and is its inconsistency with the gut flora of the common teal a lesson for common teal husbandry?

Author Response

We hope this letter finds you well. I am writing to submit the revised version of our manuscript titled “Seasonal variations over host species in shaping gut fungal and potential pathogenic communities in mixed captive Baikal Teal and Common Teal” which was initially submitted to “Animals” under the manuscript ID animals-2573308. We would like to express our gratitude to you for the insightful feedback and valuable suggestions provided during the review process.
We have carefully addressed each of your comments and suggestions in this revised version. Please find the response file in the attachment

Reviewer 4 Report

Seasonal variations over host species in shaping gut fungal and potential pathogenic communities in mixed captive Baikal teal and Common teal

Dear Authors,

the manuscript is quite well prepared and there are not many elements to correct. Mainly it is needed to check if there is not necessarily to use in case statistical analysis non-parametrical test instead of one-factorial ANOVA, and description of differences in case p < 0.05 using small letters will be important. Below I add some suggestions helpful during this process:

Line 3-4

Maybe it would be worth adding the Latin names of duck species right away in the title?

Line 52-53

Please remove the space between paragraphs.

Line 94

68 samples may give a low power of a test (0.24) in the case of using a parametric test in the form of one-factorial ANOVA. Increasing sets of data to 200 samples allows to obtain acceptable results (then 50 samples would have to be taken for each duck species in a given season, and the test power is equal 0.72). In the case of wild ducks, individual variability will be greater than in the case of livestock, where homogeneity can be increased by selecting genetically similar birds. In addition, the description of the Sample collecting 2.2 chapter does not contain information about the normality of distributions and homogeneity of variances, while the values of the medians in the charts indicate the occurrence of asymmetry of the distribution and in the case of BKS, 4 outliers are visible. Then it may be better to consider using non-parametric tests in the form of Kruskal-Wallis ANOVA or even PERMANOVA. If classic one-way ANOVA will be used, it is also important adding this information in Materials and Methods subsection along with Duncan's post-hoc test.

Line 133

A more detailed description of the four indices would be useful.

Line 163

Figure 1, significant differences are presented at level for p < 0.05. According with the scientific nomenclature better will be to present differences using small letters (a, b, c). Capital letters (A,B,C) are used in case p < 0.01.

Line 202

Figure 4. The same like in case line 163.

Line 230

Figure 6. Title should be shortened. Second sentence with description of stress values can be added in Results instead of description beneath figure.

Figure 7. The same like in case line 163.

Line 357

In case Aleppo pine, binominal nomenclature (Pinus halepensis) could be added.

Line 424-591

Abbreviations needed in case of Journal name’s, according to instruction for Authors.

Author Response

(The authors gave the same response as above.)

Round 2

Reviewer 4 Report

Dear Authors,

Thank you for the revision process. All comments and suggestions were taken into account in the cover letter and in the manuscript.

Just one little request to add two dots to the abbreviations of the first two journals in References (1. Mediators Inflam. and 2. Trends Immunol.).